# Spin diffusion from an inhomogeneous quench in an integrable system

Marko Ljubotina[1], Marko Žnidarič[1] & Tomaž Prosen[1]

Generalized hydrodynamics predicts universal ballistic transport in integrable lattice systems when prepared in generic inhomogeneous initial states. However, the ballistic contribution to transport can vanish in systems with additional discrete symmetries. Here we perform large scale numerical simulations of spin dynamics in the anisotropic Heisenberg *XXZ* spin 1/2 chain starting from an inhomogeneous mixed initial state which is symmetric with respect to a combination of spin reversal and spatial reflection. In the isotropic and easy-axis regimes we find non-ballistic spin transport which we analyse in detail in terms of scaling exponents of the transported magnetization and scaling profiles of the spin density. While in the easy-axis regime we find accurate evidence of normal diffusion, the spin transport in the isotropic case is clearly super-diffusive, with the scaling exponent very close to 2/3, but with universal scaling dynamics which obeys the diffusion equation in nonlinearly scaled time.

[1] Department of Physics, Faculty of Mathematics and Physics, University of Ljubljana, Jadranska 19, SI-1000 Ljubljana, Slovenia. Correspondence and requests for materials should be addressed to T.P. (email: tomaz.prosen@fmf.uni-lj.si).

Integrable models, such as the classical Kepler problem, harmonic oscillators, the planar Ising problem and so on, form cornerstones of our understanding of nature. Their equilibrium physics is usually well understood, even for the most complicated among integrable models, for example, the ones solvable by the Bethe ansatz[1]. Non-equilibrium physics of quantum systems on the other hand is much less understood[2], particularly when going beyond the simplest integrability of quadratic models. This theoretical gap is becoming even more apparent with the advancement of experimental methods that are offering us analogue simulation of models beyond the capability of our best theoretical and numerical methods[3,4].

Non-equilibrium dynamics of integrable quantum systems is thus one of the main current focuses of both theoretical and experimental condensed matter physics[5]. A macroscopic number of conservation laws existing in such systems[6] provide a variety of ways to break ergodicity, manifesting, for instance, in equilibration processes to non-thermal states or ballistic high-temperature transport of conserved quantities, such as energy, magnetization or charge. A naive classical reasoning might be that, because integrable systems are distinguished by constants of motion that force the dynamics to be simple and almost periodic (for example, orbits winding up the torus), one should expect to see ballistic transport. We shall demonstrate that this picture, while being correct for trivially integrable noninteracting models, such as harmonic oscillator chains[7], can in fact be wrong for an interacting quantum integrable model.

Recently, a generalization of hydrodynamics has been put forward[8,9] which successfully predicts ballistic currents and scaled density profiles of integrable interacting systems quenched from inhomogeneous initial states[10–15], which is a convenient method to study relaxation and non-equilibrium transport. In this protocol, the system is prepared in the state where the left and the right part, for $x < 0$ and $x > 0$, respectively, are in different equilibrium states, and then, at $t = 0$, let to evolve with a homogeneous interacting Hamiltonian. However, when ballistic transport is prohibited due to generic symmetries, such as is the case for spin transport in the anisotropic Heisenberg spin chain in the easy-axis (Ising) regime, this theory makes no prediction.

In extended interacting integrable system a macroscopic number of local conservation laws exists, in number proportional to the number of degrees of freedom, which can be exploited to develop generalized hydrodynamics[8]. This theory for typical inhomogeneous initial states predicts ballistic scaling $f(\xi = x/t)$ of densities and currents of conserved quantities, such as energy, charge or magnetization. However, in systems with parity ($\mathbb{Z}_2$) symmetries, such as particle-hole exchange (or spin reversal), and for observables that are odd under the parity and initial states that are symmetric under the combined parity and spatial reflection $x \rightarrow -x$, the ballistic contribution to transport can vanish. In fact, vanishing ballistic transport channel can then be related to the absence of local or quasi-local conserved charges with odd parity[6,16]. This means that the transported conserved quantity at $x = 0$ grows slower than linear with $t$.

Here we propose a conjecture, based on large scale simulations, that a quench from an inhomogeneous initial state will in such cases generically result in diffusive spin dynamics. We demonstrate our results on the anisotropic Heisenberg chain (XXZ model). However, we stress that the XXZ model goes beyond being a mere toy model—it has been instrumental in the development of quantum integrability[17,18] and describes interaction in real spin chain materials[19]. Remarkably, in the case of isotropic Heisenberg interaction, spin relaxation is super-diffusive but with universal scaling dynamics which obey the standard diffusion equation in nonlinearly scaled time. Our results thus reveal a surprising property of an important

integrable model as well as pose a challenge to theories which at present are unable to account for our observations. Because the parity symmetry is ubiquitous, our set-up should be widely applicable, for instance, we predict a similar physics in the one-dimensional (1D) Hubbard model.

## Results

**The set-up.** The Hamiltonian of the XXZ chain of $n$ sites reads

$$\mathcal{H} = J \sum_{x=-n/2}^{n/2-1} \left( s_x^{(1)} s_{x+1}^{(1)} + s_x^{(2)} s_{x+1}^{(2)} + \Delta s_x^{(3)} s_{x+1}^{(3)} \right), \quad (1)$$

where $\Delta$ is the anisotropy parameter and $s_k^{(\gamma)} = \frac{1}{2} \sigma_k^{(\gamma)}$ are the spin 1/2 operators, with Cartesian component $\gamma = 1, 2, 3$, expressed in term of Pauli matrices $\sigma_k^{(\gamma)}$ (we use units $J = \hbar = 1$). The Hamiltonian preserves the total magnetization, $M = \sum_x s_x^{(3)}$, $[\mathcal{H}, M] = 0$. We are going to study the spin transport satisfying the continuity equation $ds_x^{(3)}/dt = j_{x-1} - j_x \approx -\nabla j_x$ with the current

$$j_x = s_x^{(1)} s_{x+1}^{(2)} - s_x^{(2)} s_{x+1}^{(1)}. \quad (2)$$

The existence of spin-reversal parity $S = \prod_x \sigma_x^{(1)}$, $[\mathcal{H}, S] = 0$ and odd current $j_x S = -S j_x$, implies an absence of ballistic transport channels based on local conserved charges[20]. We are going to simulate the time evolution of an initial inhomogeneous state composed of two halves with opposite magnetizations.

To this end we choose a product initial state described by a density operator $\rho$,

$$\rho(t=0) \sim \left( 1 + \mu \sigma^{(3)} \right)^{\otimes \frac{n}{2}} \otimes \left( 1 - \mu \sigma^{(3)} \right)^{\otimes \frac{n}{2}}, \quad (3)$$

where the parameter $\mu \in [-1, 1]$ determines the initial magnetization, being $\langle s_{x \geq 0, < 0}^{(3)} \rangle = \pm \frac{1}{2} \mu$. Each of the initial halves can be thought of as being in equilibrium state $\sim e^{\pm h \sum_x s_x^{(3)}}$ at very high temperature and finite magnetization. We are therefore studying high-energy non-equilibrium physics of the model. While the initial state is pure for $|\mu| = 1$ (a fully polarized domain wall), evolution of which has been studied in the past[21], the choice of a mixed state offers several important advantages: it is generic and not plagued by the speciality of $\mu = 1$ at $\Delta > 1$ for which the dynamics freezes due to the proximity to a gapped eigenstate[22], and it is, for small $\mu$, better suited for numerical simulations. This allows us to study significantly longer timescales as compared to existing literature and infer the scaling functions. We also mention that such an initial state can be thought of as representing an ensemble of pure states with randomized angle $\varphi$ on the Bloch sphere (Methods section).

**Scaling exponents.** We focus our efforts on $\Delta \geq 1$ where there are no analytic results known for the magnetization transport, and the method[8,9] only predicts vanishing ballistic contribution. Two representative examples of a time evolved state $\rho(t)$, namely the spin and current profiles $s(x, t) = \mathrm{tr} \rho(t) s_x^{(3)}$, $j(x, t) = \mathrm{tr} \rho(t) j_x$, are shown in Fig. 1. To obtain the exact type of transport we shall quantitatively study equilibration of magnetization, in particular the scaling of spin and current profiles as well as the transferred magnetization between the two halves, whose asymptotic scaling power $\alpha$ characterizes the transport type,

$$\Delta s(t) = \int_0^t j(0, t') dt' \propto t^\alpha, \quad (4)$$

where $j(0, t)$ is the current at the half-cut. For $\alpha = 1/2$ the transport is diffusive, for $1/2 < \alpha < 1$ it is called super-diffusive, and finally, $\alpha = 1$ corresponds to ballistic transport. We note that the transport type is connected to current–current correlation

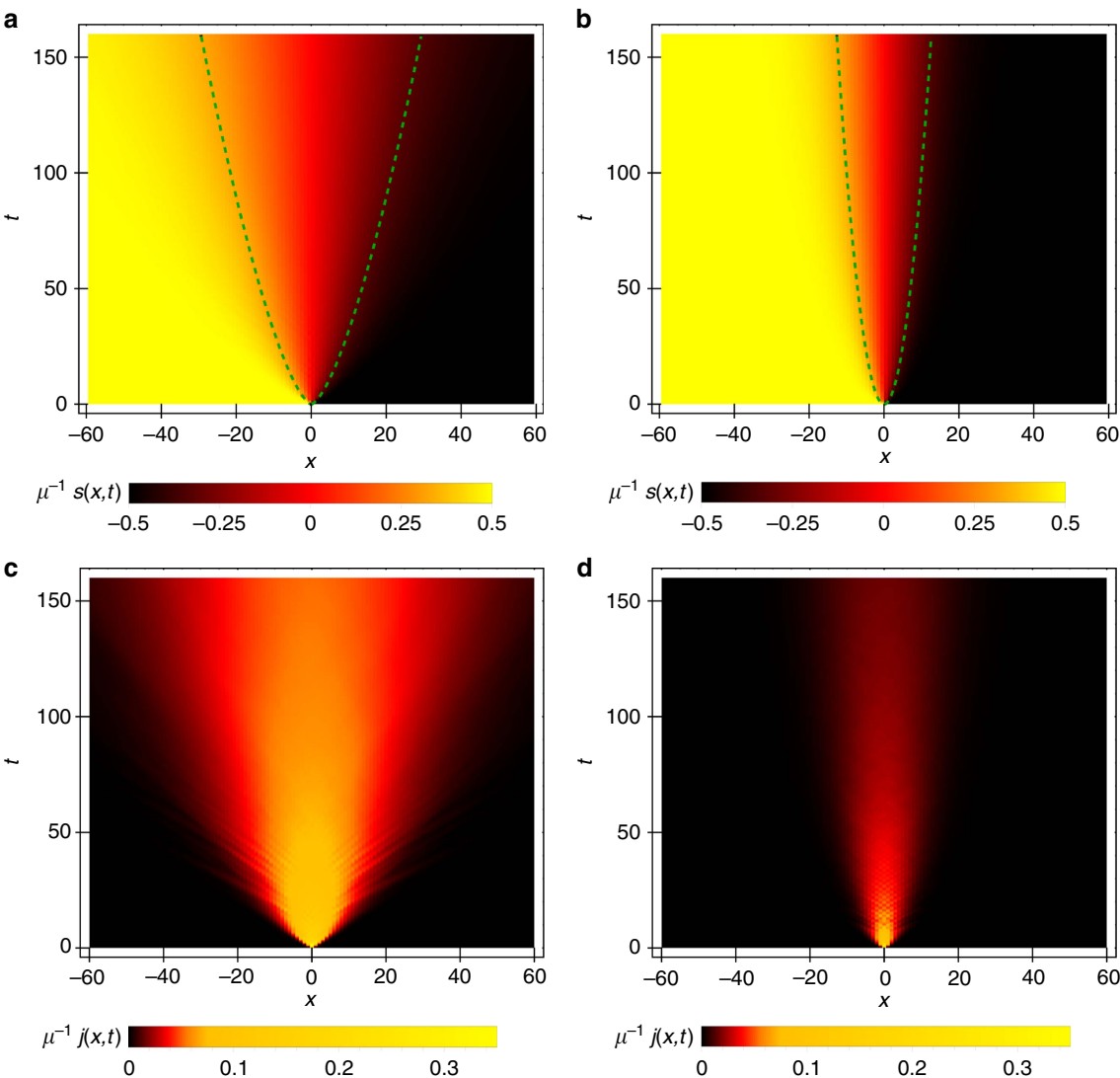

**Figure 1 | Dynamics of spin and current densities.** Time evolution of spin density $s(x, t) = \mathrm{tr}(\rho(t)s_x^{(3)})$ (**a,b**) and current (**c,d**) profile $j(x, t) = \mathrm{tr}(\rho(t)j_x)$ for the isotropic point $\Delta = 1$ (**a,c**), and $\Delta = 2$ (**b,d**), following an inhomogeneous quench. One can see that the spreading is much faster for $\Delta = 1$, in both cases though it is slower than ballistic. Dashed green curves guide the eye towards scaling $x \sim t^{2/3}$ in **a**, and $x \sim t^{1/2}$ in (**b**). Data are shown for $n = 320$ and small initial polarization $\mu = \pi/1{,}800$.

function via Green–Kubo linear response theory. In case of diffusive transport, the spin density satisfies the diffusion equation. This notion of diffusion does not necessarily correspond to De Gennes phenomenological theory of spin diffusion which, under much stronger assumptions, in 1D implies $1/\sqrt{t}$ dependence of local spin density autocorrelation function[23,24].

We evolved the initial state $\rho(0)$ (3) up to long times (of order $t \approx 160$) and set large enough $n$ so that there was no significant finite size effects. From the data we then infer the exponent $\alpha$ using equation (4), see Fig. 2a,b for representative plots. Dependence of the exponent $\alpha$ on $\Delta$ is summarized in Fig. 2c. While the transport is found to be ballistic for $\Delta < 1$, expectedly so for the integrable system, also known rigorously[16], at $\Delta \geq 1$ we find rather clear non-ballistic relaxation. In particular, at $\Delta = 1$ it is super-diffusive while for $\Delta > 1$ the transport is diffusive, observed in driven steady-state setting[25,26] as well as in the Hamiltonian one[24,27–30]. At $\Delta = 1$ we also observe small dependence of $\alpha$ on $\mu$. While for small $\mu$, that is, small deviations

from an infinite temperature state $\rho \sim \mathbb{1}$, the exponent is close to 2/3, closer to pure state $\mu = 1$ it appears to be closer to $\approx 3/5$ (we note that a different numerical procedure is used in the two regimes, see Methods).

**Scaling functions**. The scaling of the transferred magnetization unequivocally shows a surprising non-ballistic transport in an integrable system which, however, has been observed and discussed before in related contexts, namely within local quench and linear response theory[24,27–30] and boundary driven Lindblad approach[25,26]. But here we can do still more. In Fig. 3 we demonstrate that the spin profiles can be described by a function of a single-scaling variable $x/t^\alpha$—profiles at large times collapse to a single curve. In addition, the profiles of current and magnetization are proportional to each other at different times (Fig. 3c,d), therefore validating Fick's law $j = -D\nabla s$ where the behaviour of the diffusion constant $D$ with respect to the anisotropy $\Delta$ is shown in the inset of Fig. 2c. This comes as no

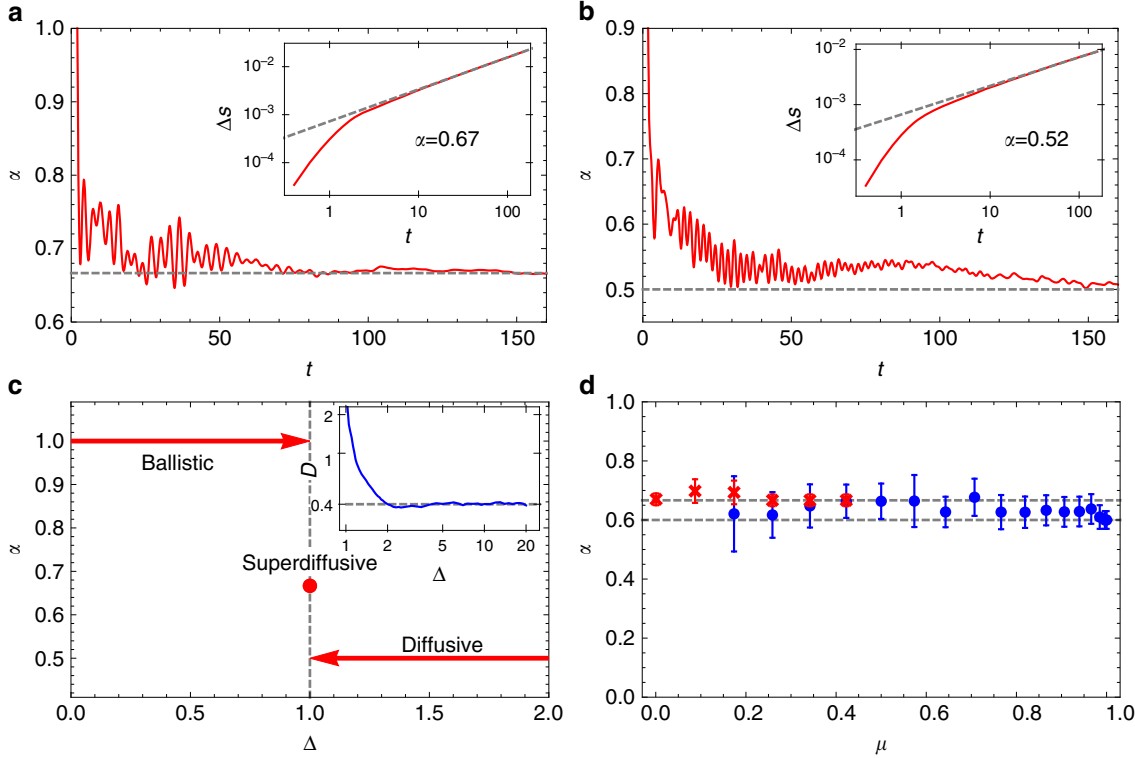

**Figure 2 | Scaling exponents of magnetization spreading.** (**a,b**) Local exponent $\alpha(t)$ calculated as a numerical log-derivative $d \log \Delta s(t)/d \log t$ for $\Delta = 1$ (**a**) and $\Delta = 2$ (**b**) (dashed lines indicate exponents 2/3 and 1/2, respectively, while dashed lines in the insets show best power-law fits to $\Delta s(t)$—red curve), both for $\mu = \pi/1,800$. (**c**) Conjecture for the dependence $\alpha(\Delta)$ at high temperatures and small $\mu$. The inset shows the diffusion constant obtained from Fick's law for various values of $\Delta$ in the diffusive regime, converging to a finite value at large $\Delta$ (agreeing with ref. 28). (**d**) Dependence on $\mu$ for $\Delta = 1$ shows a small but significant change in the behaviour: for $\mu \approx 1$ it is closer to $\alpha = 3/5$ while for small $\mu$ it becomes close to $\alpha = 2/3$ (dashed). The blue (circles) and red (crosses) symbols represent wave function and density operator evolutions respectively. We average over samples of 10–130 random initial wave-functions for each blue data point. For intermediate $\mu$ the error-bars (denoting the estimated s.d.) are larger since the simulation is less efficient in that regime (Methods section).

surprise in the diffusive regime $\Delta > 1$ where the scaling function of the magnetization (Fig. 3b) is simply the error function $s(x,t) = -\frac{\mu}{2} \mathrm{erf}(x/\sqrt{4Dt})$. However, the same can not be said for the isotropic point $\Delta = 1$. Proportionality between the magnetization gradient and the current profile (Fig. 3c), this time with a time-dependent ratio $D \simeq \frac{K}{3} t^{1/3}$, suggests a diffusion equation in a scaled time

$$\frac{\partial s(x,t)}{\partial \tau} = \frac{K}{4} \frac{\partial^2 s(x,t)}{\partial x^2}, \quad \text{where} \quad \tau = t^{4/3}, \qquad (5)$$

which again yields error function profile with a different scaling variable $s(x,t) = -\frac{\mu}{2} \mathrm{erf}(K^{-1/2} x/t^{2/3})$ with $K = 2.33 \pm 0.03$. In Fig. 3a we compare numerical profiles with the error function, again finding good agreement within accuracy of our simulations. Therefore, the scaling function is, in both cases, $\Delta = 1$ and $\Delta > 1$, the error function, the difference being only in the scaling variable which is $x/t^{2/3}$ at the super-diffusive isotropic point. This result is surprising, as anomalous diffusion is usually associated with Levy processes and hence long (non-Gaussian) tails in the profiles. Here it seems it all amounts to a nonlinear rescaling of time. Theoretical explanation of this effect is urgent.

**Entanglement entropy and simulation complexity.** Finally, we mention a numerical observation that explains why we can simulate dynamics to such long times, and is an interesting property on its own. We use a time-dependent density matrix renormalization group method (tDMRG), see Methods.

The efficiency of tDMRG depends on the entanglement entropy, that is, for pure state evolution on the Von Neumann entropy $S = -\mathrm{tr}[\rho_A \ln \rho_A]$ of the reduced state $\rho_A = \mathrm{tr}_A|\Psi\rangle\langle\Psi|$, whereas for mixed states evolution on an analogous operator space entanglement entropy $S^\#$, (ref. 31) of a vectorised density operator $\rho$. When starting with a typical product initial state both entropies typically grow linearly with time, regardless of the system being integrable or not[32,33], causing exponentially fast growth of complexity and with it a failure of these numerical methods. In our case though, see Fig. 4, entropies grow much slower, namely in a power-law fashion

$$S \sim t^\beta, \quad \text{or} \quad S^\# \sim t^\beta, \qquad (6)$$

with $\beta$ being $<1$. The most efficient simulations have been possible with density operators for small $\mu$ where the exponent $\beta$ is typically between 0.3 and 0.5.

## Discussion

Our numerical results can be interpreted as an evidence of normal spin diffusion and spin Fick's law in the easy-axis anisotropic Heisenberg chain (for anisotropy $\Delta > 1$), with spin density satisfying the diffusion equation on large scales. Besides the case $\Delta = 2$ shown here, we provide additional data for $\Delta = 1.05, 1.1, 1.3, 1.5$ demonstrating a clear convergence of the diffusive scaling exponents $\alpha = 1/2$ in all massive cases (Supplementary Note 1), and data for massless cases $\Delta = 0, 0.5, 0.7, 0.9$ which indicate convergence to ballistic exponent $\alpha = 1$ (Supplementary Note 2).

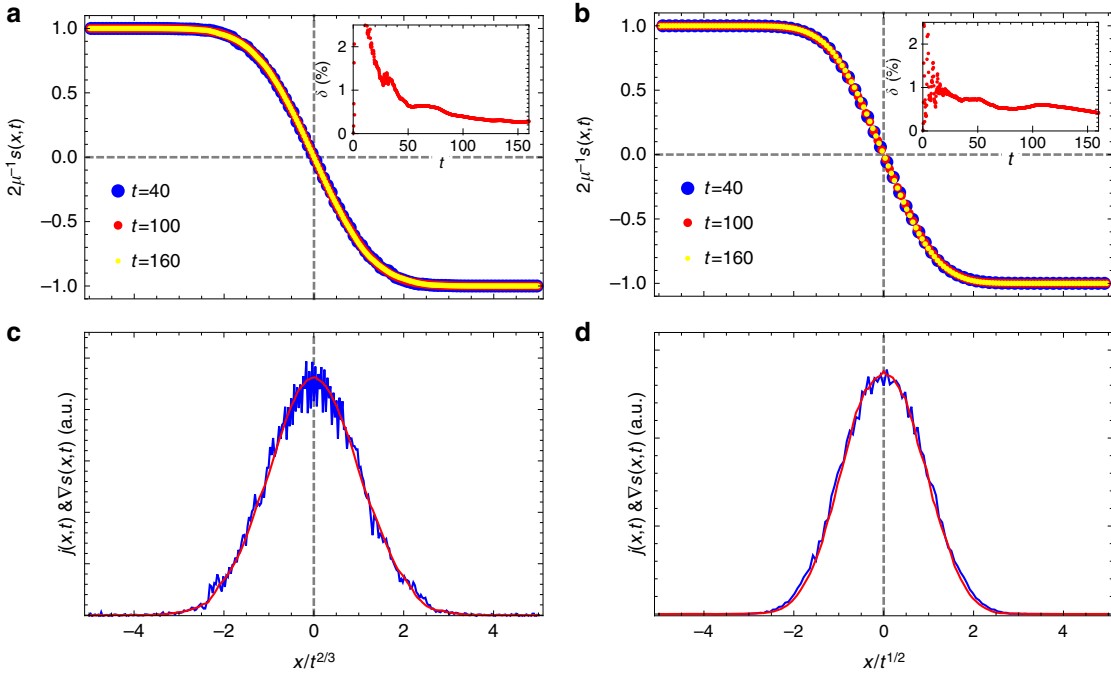

**Figure 3 | Scaling profiles.** Scaling of density and current profiles with $x/t^\alpha$. In (**a,b**) we show the scaling of magnetization profiles, (**a**) for $\Delta=1$ using $\alpha=2/3$, and (**b**) for $\Delta=2$ and using $\alpha=1/2$ (note that the points for different times overlap almost perfectly; the insets show the convergence of the relative root-mean-square difference (in %) between data $s(x,t)$ and scaled erf-profiles (see text) as a function of time). Frames (**c,d**) show the emergence of Fick's law at late times (shown at $t=160$), comparing current profiles (red) to gradients of spin density (blue)—both indistinguishable from Gaussians, for $\Delta=1$ in (**c**) and $\Delta=2$ in (**d**). In all plots the system size is $n=320$.

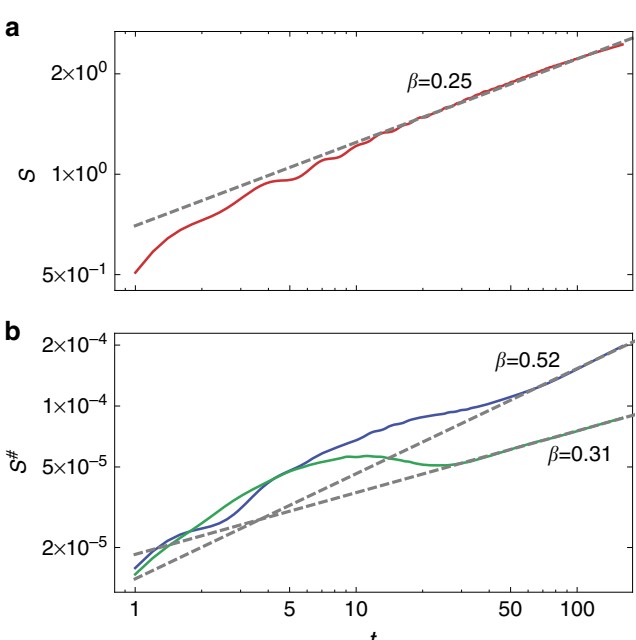

**Figure 4 | Simulation complexity.** (**a**) Von Neumann entanglement entropy $S$ for the fully polarized initial state ($\mu=1$) at the isotropic point $\Delta=1$. (**b**) Operator space entanglement entropy $S^\#$ for $\Delta=1$ (blue) and $\Delta=2$ (green), both for $\mu=\pi/1,800$. Bipartition into two equal halves and a system size of $n=320$ are used.

While for generic, non-spin-reversal-symmetric initial states, the dominant contribution to transport is ballistic as determined by generalized hydrodynamics (or generalized 1D Euler's equations)[8,9,13–15], the next-to-leading term is now clearly

predicted to be diffusive, as following from our work. However, a theoretical explanation, or even derivation of a diffusive contribution to transport in an integrable system with a macroscopic number of conservation laws is still pending. Even more surprising is the discovery of anomalous super-diffusive transport in the isotropic case ($\Delta=1$) with the scaling exponent equal to or very close to 2/3. While this might suggest a behaviour described by KPZ (Kardar–Parisi–Zhang) universality class, we find that asymptotic spin density profiles obey the nonlinearly scaled diffusion equation and are distinct from the KPZ profiles. One might conjecture that the scaling exponent 2/3 is a consequence of $SU(2)$ symmetry and not the fact that the model there corresponds to the marginal critical point $\Delta=1$. This would be consistent with observed anomalous super-diffusive scalings in $SU(4)$ spin ladders in the set-up of driven steady-state Lindblad dynamics[34] where the scaling exponent appears to be $\alpha=3/5$. Curiously, all scaling exponents observed in this work (1/1, 1/2, 2/3, 3/5) are ratios of subsequent Fibonacci numbers[35].

## Methods

**Numerical procedures.** The time evolution is performed by means of the tDMRG algorithm[36,37]. In particular, for small $\mu$ data (which is mostly reported here) the most efficient was the matrix product density operator version of tDMRG, with which we could reach times of the order $t\simeq200$ for system size $n\simeq2t$ using bond dimensions 50–200 resulting in relative truncation errors $<1\%$. One the other hand, for $\mu\approx1$ (close to domain wall pure state), the pure state version of tDMRG becomes more efficient as the corresponding entanglement entropy scaling exponents $\beta$ are smaller. The two approaches appear to complement one another as can be seen in Fig. 2d. Neither approach allows us to observe long times in the intermediate region of $\mu$, where the exponents $\beta$ become closer to 1.

In order to simulate the desired density operator by evolving pure states we define a set of initial states

$$|\Psi(t=0)\rangle=\bigotimes_{x<0}|\psi(\mu,\phi_x)\rangle\otimes\bigotimes_{x\geq0}|\psi(-\mu,\phi_x)\rangle \quad (7)$$

where $|\psi(\mu,\phi)\rangle=\sqrt{(1+\mu)/2}|\uparrow\rangle+e^{i\phi}\sqrt{(1-\mu)/2}|\downarrow\rangle$ is simply the Bloch sphere representation of a 2-level system and the $\phi_x$ are uniform independent random

numbers in the range $[0, 2\pi]$. The density matrix is then obtained as an ensemble average over a set of such pure random states $\rho(t) = \mathbb{E}(|\Psi(t)\rangle\langle\Psi(t)|)$. It is clear that an increasingly large set of random states is needed as the magnetization approaches $\mu \to 0$, where the matrix product density operator simulation is favourable anyway.

**Data availability.** Data are available on request from the authors.

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

## Acknowledgements

We would like to acknowledge support from ERC grant OMNES, as well as grants J1-7279, N1-0025 and P1-0044 of the Slovenian Research Agency.

## Author contributions

T.P. proposed the project, M.L. performed the simulations and all authors contributed to the analysis and writing of the manuscript.

## Additional information

**Competing interests:** The authors declare no competing financial interests.

**Publisher's note**: 

