## [Peer Review File · Nature Communications]

Reviewers' comments:

Reviewer #1 (Remarks to the Author):

In this paper, the authors report on results of a large-scale numerical calculation of spin current and density in the XXZ quantum spin chain out of equilibrium. The setup is a relatively standard protocol, where an initially prepared imbalance is let to evolve unitarily, and the transfer of various quantities - energy, particles, spin - is observed. This paper addresses in particular the question of spin transfer in non-ballistic regimes. The main point is to determine with high numerical accuracy the form of the non-ballistic transport of spin in such regimes: diffusive and super-diffusive. This paper does not purport to provide full explanations, but rather to give strong numerical evidence, and so I will judge the paper mainly upon this basis.

Three questions which I would like the authors to clarify in their manuscript, some which are more a matter of style and presentation (but are important):

1) The argument presented in the first page is that observables that are odd under parity with initial state invariant under parity + space inversion, lead to vanishing of ballistic current. Of course, the current operator for a density that is odd under parity, is itself even under parity + space inversion, and thus, as I understand, there is no immediate symmetry argument to imply absence of ballistic transport. This is not the true reason of course - the full argument has to do with the set of conserved charges, and the absence / presence of charges with appropriate symmetry transformation. In this sense, I'm not sure the right-hand paragraph on p1 is a good representation of the situation. Also, then, the ubiquity might be more complicated than what appears to be suggested here. It would be good if the authors could clarify the phrasing - and perhaps explain a bit more how general they believe their main results should be (in what other models should similar results occur?).

2) The non-ballistic transport is portrayed as surprising (e.g. p 4, left). It is of course interesting in integrable models, but well established in the literature, as the authors correctly cite. Is non-ballistic transport observed by the author actually new? Perhaps the particular set of initial states they consider was never considered before (although similar states were considered)? Why is non-ballistic transport still surprising, given that it has been explained? This is perhaps just a matter of style, and I agree that it is a good idea to point out this fact to a wide audience, but again I would be happy if the authors could re-phrase a little bit in order to represent the current situation (unless I am misunderstanding it).

3) The fact that $\Delta=1$ is particular is quite interesting. In the interest of a general understanding and in view of possible generalizations to other models, it would be nice if the authors could provide some intuition as to the properties of this point that makes it special with respect to transport (a boundary point of a critical line? is this its important feature? what other feature of this point may lead to the observed behaviours? the SU(2) symmetry?). I know the authors might not have full understanding, but maybe some intuition from their numerical experiment.

Points that are interesting:

1) the precise observation of, not only non-ballistic, but diffusive behaviour at $\Delta>1$ is interesting, although, as cited by the authors, observed previously in similar situations. The precise form of the current and density is interesting, for future analytic studies to reproduce.

2) the observation of super-diffusion at $\Delta=1$, with μ -dependent exponent, is a new discovery, certainly demanding a proper theoretical explanation. I think this provides good results at the basis of further studies, and thus can foster new research.

3) the observation that Fick's law is satisfied both at $\Delta > 1$ and $\Delta = 1$ is interesting, especially in the latter case. It is a clear numerical result necessitating a theoretical explanation.

4) the observation, at $\Delta = 1$, of the collapse to an error function upon appropriate scaling is definitely very surprising, and of great interest.

The numerical evidence presented I find sufficiently convincing. I believe points 2, 3, 4 are very likely to foster new research on the subject, and are key, nontrivial numerical observations in the context of quantum transport. Hence, after the modifications above are made, I think it is appropriate for publication.

Reviewer #2 (Remarks to the Author):

Integrable systems count with an extensive number of local conserved quantities, which allow one to make predictions concerning their thermodynamic and non-equilibrium behavior. Recent developments have been instrumental at generalizing statistical mechanics to quantum integrable problems, but numerous questions remain unanswered. Besides the extensively studied problem of thermalization, there are important issues regarding their non-equilibrium behavior, particularly after a quench or applied bias. In certain systems, starting from typical inhomogeneous states, a recently developed theory predicts ballistic scaling, which is the conventional assumption in the case of integrable models. But this theory makes no predictions when additional symmetries are present.

This work addresses this issue for a paradigmatic integrable model (Heisenberg chain), starting from an inhomogeneous mixed state, mean-field like, that can be described as a chain at high temperature with half of the system being under the action of a magnetic field in one direction, the other half under a magnetic field in the opposite direction. The time evolution calculations are simple and straight-forward, and I have no reason to doubt the accuracy of the results, that show that transport is diffusive for the gapped regime ($\Delta > 1$) and, even more surprisingly, super diffusive at the SU(2) symmetric point ($\Delta = 1$). In addition, they demonstrate that in the super diffusive regime the magnetization profile obeys a scaling law in terms of a single scaling variable α , without any free fitting parameters.

Even though the setup and calculations are fairly simple, the authors were able to identify a behavior that defies all previous assumptions about the transport dynamics in integrable systems. I suspect the more we move away from simple trivial examples with trivial initial states, we might encounter more of such counter examples. These results are definitely very surprising, and will stimulate a great deal of effervescence in the field. I am sure the community will be greatly motivated by this work, and we will witness new developments in the theory of integrable systems. I therefore recommend it for publication.

Two remarks:

- The authors could cite Phys. Rev. A 93, 021607(R) (2016), which I consider relevant in the context of this work.

- The authors could supply supplementary data showing that indeed their calculations support ballistic behavior for $\Delta < 1$.

Reviewer #3 (Remarks to the Author):

The authors study the evolution of spin and density profiles in the integrable XXZ chain following an inhomogeneous quench. The results at the isotropic point, anisotropy $\Delta = 1$, and in the massive regime, anisotropy $\Delta > 1$, are obtained using a numerical method based on a matrix product representation of states and operators.

Nonequilibrium dynamics in integrable one-dimensional models and transport in particular, are very active fields of research in condensed matter and ultracold quantum gases. Considerable progress has been made in recent years showing that interacting Bethe ansatz integrable models display much more complex dynamics than non-interacting quadratic Hamiltonians.

The topic of the paper, spin diffusion in the XXZ chain, has already been investigated previously. In particular, a similar setup and method has been used in Ref. [23] of the manuscript where also similar results for the spin transport in the massive regime have been reported. However, the time scales over which the dynamics has been simulated by the authors are an order of magnitude larger than in Ref. [23]. This allows to perform for the first time a detailed scaling analysis of the density profiles. The authors provide, in particular, strong evidence for a rather unusual superdiffusive behaviour at the isotropic point described by Eq. (5) in the manuscript. This is the main novel result which, I believe, constitutes significant progress in this field of research.

While the paper, in my view, thus fulfills all the general criteria to be published in Nat. Comm. I also do have specific concerns which need to be addressed:

1) Spin diffusion is perhaps most commonly understood as a $1/t^{1/2}$ scaling of the spin-spin autocorrelation function going back to de Gennes phenomenological theory. Spin diffusion in spin chains defined in this way has a long history and has been studied theoretically and experimentally (NMR). The authors, on the other hand, consider diffusive behavior in the spin current. This distinction should be made clear in the introduction and appropriate references should be cited. It should be clarified that the spin-spin autocorrelation function does show diffusive behavior at finite temperatures also in the gapless regime (see e.g. PRB 57, 8340 (98) and several more recent works). Without such a clarification the paper might - at least at first glance - be quite confusing for non-experts.

2) I am also concerned that the authors make claims which are stronger than what the data really support. Looking at Fig. 2(b) it looks to me as if the data for $\Delta=2$ show an effective exponent $\alpha > 1/2$ up to times $t \sim 100$ before the exponent gets closer to $1/2$ at the longest times they can simulate. Is it possible that for Δ values closer to 1 the numerics shows an exponent larger than $1/2$ for all accessible times? If this is the case then the authors seem to interpret this as a crossover to the true exponent perhaps with a diverging time scale when approaching $\Delta=1$. It seems to me that an alternative interpretation with an exponent α which depends continuously on Δ might not be excluded by the data. This point should be discussed. Showing at least one data set for a Δ value closer to 1 would be helpful.

3) Caption of Fig. 1, last line: the scaling seems to be interchanged, i.e., it probably should read $x \sim t^{1/2}$ for (b).

Response to reviewers' comments:

Reviewer #1:

We thank the referee for her/his careful consideration of our manuscript and for favourable recommendation. We considered all the questions and comments to which we reply below.

Question:

1) The argument presented in the first page is that observables that are odd under parity with initial state invariant under parity + space inversion, lead to vanishing of ballistic current. Of course, the current operator for a density that is odd under parity, is itself even under parity + space inversion, and thus, as I understand, there is no immediate symmetry argument to imply absence of ballistic transport. This is not the true reason of course - the full argument has to do with the set of conserved charges, and the absence / presence of charges with appropriate symmetry transformation. In this sense, I'm not sure the right-

hand paragraph on p1 is a good representation of the situation. Also, then, the ubiquity might be more complicated than what appears to be suggested here. It would be good if the authors could clarify the phrasing - and perhaps explain a bit more how general they believe their main results should be (in what other models should similar results occur?).

Answer:

We have provided a more detailed discussion of our setup in the revised introduction, bringing up also the connection between ballistic transport and the existence of parity-odd (quasi)local conserved charges. Additionally, we discuss now the generality of our results in the last sentence of introduction, where we mention the 1D Hubbard model as another important candidate model where similar physics should be observed.

Question:

2) The non-ballistic transport is portrayed as surprising (e.g. p 4, left). It is of course interesting in integrable models, but well established in the literature, as the authors correctly cite. Is non-ballistic transport observed by the author actually new? Perhaps the particular set of initial states they consider was never considered before (although similar states were considered)? Why is non-ballistic transport still surprising, given that it has been explained? This is perhaps just a matter of style, and I agree that it is a good idea to point out this fact to a wide audience, but again I would be happy if the authors could rephrase a little bit in order to represent the current situation (unless I am misunderstanding it).

Answer:

We fully agree with the referee. Even though the appearance of diffusion in integrable systems could still appear surprising to many theoretical physicists, it has been pointed out and discussed before (in particular in the references that we mention). We have emphasised this by quoting the relevant references in the first place that we mention observation of non-ballistic transport (in the beginning of the paragraph on “scaling functions”) and spell out the contexts in which it has been discussed. We have as well added a new related reference here (Steinigeweg et al PRB 95, 035155 (2017)) which we have missed in the first version of the manuscript.

Question:

3) The fact that $\Delta=1$ is particular is quite interesting. In the interest of a general understanding and in view of possible generalizations to other models, it would be nice if the authors could provide some intuition as to the properties of this point that makes it special with respect to transport (a boundary point of a critical line? is this its important feature? what other feature of this point may lead to the observed behaviours? the SU(2) symmetry?). I know the authors might not have full understanding, but maybe some intuition from their numerical experiment.

Answer:

We discuss this interesting point at the end of the new “Discussion” section. Of course, we can only provide speculative thoughts at this point.

Reviewer #2:

We thank the referee for her/his careful consideration of our manuscript and for favourable recommendation. We have carefully considered the remarks of the referee and provide our answers below.

Remark:

- The authors could cite Phys. Rev. A 93, 021607(R) (2016), which I consider relevant in the context of this work.

We have studied the reference mentioned by the referee, but after careful consideration decided not to include it in our paper. Namely, the reference discussed ballistic expansions of integrable systems (local quenches), but our work focuses on non-ballistic transport and inhomogeneous (global) quenches. As there are many other relevant papers on ballistic local quenches, we feel there would be no justifiable reason to mention just that particular reference. We hope that the referee is not categorical on this point.

- The authors could supply supplementary data showing that indeed their calculations support ballistic behaviour for $\Delta < 1$.

We have now added Supplementary Information document to our manuscript, and in Supplementary Note 2 we demonstrate clearly that our simulations reproduce ballistic behavior for $\Delta < 1$. For non-interacting case ($\Delta = 0$) we even find excellent agreement with known analytic solutions.

Reviewer #3:

We thank the referee for her/his careful consideration of our manuscript and for favourable recommendation. We considered all the questions and comments to which we reply below.

Question:

1) Spin diffusion is perhaps most commonly understood as a $1/t^{1/2}$ scaling of the spin-spin autocorrelation function going back to de Gennes phenomenological theory. Spin diffusion in spin chains defined in this way has a long history and has been studied theoretically and experimentally (NMR). The authors, on the other hand, consider diffusive behavior in the spin current. This distinction should be made clear in the introduction and appropriate references should be cited. It should be clarified that the spin-spin autocorrelation function does show diffusive behavior at finite temperatures also in the gapless regime (see e.g. PRB 57, 8340 (98) and several more recent works). Without such a clarification the paper might - at least at first glance - be quite confusing for non-experts.

Answer:

We have added three explanatory sentences in the paragraph after Eq. (4) in order to avoid confusion with the De Gennes phenomenological theory of spin-diffusion. There we also quoted the mentioned reference by Fabricius and McCoy.

Question:

2) I am also concerned that the authors make claims which are stronger than what the data really support. Looking at Fig. 2(b) it looks to me as if the data for $\Delta=2$ show an effective exponent $\alpha > 1/2$ up to times $t \sim 100$ before the exponent gets closer to $1/2$ at the longest times they can simulate. Is it possible that for Δ values closer to 1 the numerics shows an exponent larger than $1/2$ for all accessible times? If this is the case then the authors seem to interpret this as a crossover to the true exponent perhaps with a diverging time scale when approaching $\Delta=1$. It seems to me that an alternative interpretation with an exponent α which depends continuously on Δ might not be excluded by the data. This point should be discussed. Showing at least one data set for a Δ value closer to 1 would be helpful.

Answer:

We thank the referee for this question. We have now provided Supplementary Information material where in Supplementary Note 1 we show data for several different values of anisotropy in the massive regime. We see very clear convergence towards a universal diffusive exponent $1/2$ even for much smaller Δ (up to $\Delta=1.1$), and even for $\Delta=1.05$ we find a clear trend from which one could extrapolate the diffusive exponent. We believe that these data strongly corroborate our interpretation on asymptotic behaviour.

Question:

3) Caption of Fig. 1, last line: the scaling seems to be interchanged, i.e., it probably should read $x \sim t^{1/2}$ for (b).

Answer:

We thank the referee for pointing out this typo. It has been corrected.

REVIEWERS' COMMENTS:

Reviewer #1 (Remarks to the Author):

I am happy with the changes made. I believe the elements of discussion are very instructive. I think the paper can be published as it is.

Reviewer #3 (Remarks to the Author):

The authors have addressed my two main points (discussion of phenomenological spin diffusion and additional data for anisotropies close to 1). I am satisfied with the changes and recommend the manuscript for publication.